# Owner Perceived Behavior in Cats and the Influence of Husbandry Practices, Housing and Owner Attitudes in Sweden

**DOI:** 10.3390/ani12192664

**Published:** 2022-10-04

**Authors:** Elin N. Hirsch, Johanna Geijer, Maria Andersson

**Affiliations:** Department of Animal Environment and Health, Swedish University of Agricultural Sciences, P.O. Box 234, 53223 Skara, Sweden

**Keywords:** behavior problems, cat–human relationship, domestic cat, husbandry, outdoor access

## Abstract

**Simple Summary:**

In this paper, we describe the results from an online survey that targeted cat owners in Sweden. The aim of the study was to investigate how owners perceived the behavior of their cats, and if the cat characteristics could influence the owner attitudes. In this study, we describe the data from 3253 cats and their owners. Less than one in five cat owners reported that they had experienced behavioral problems with their cats. Owners of cats with outdoor access reported experiencing fewer problems. However, we could not find an effect of the length of time a cat was left home alone. If owners had university level knowledge in animal behavior, we could see an effect on the perception of cat behaviors such as cats misbehaving out of spite. Many owners did not believe that cats could be trained to overcome behavior problems, and many owners also thought that cats could manage on their own. These attitudes can in the long run have negative effects on the welfare of cats.

**Abstract:**

This study consisted of an online survey based on a convenience sample among cat owners in Sweden. The aim was to investigate how owner and cat characteristics influenced the perceived behavior of cats, focusing on perceived behavioral or temperamental problems. The relation between owner knowledge, the provided environment, and owner perceived behavior of 3253 pet cats were investigated. Few respondents (18%) reported behavioral or temperamental problems, and consequently 82% perceived no problem whatsoever. Fewer cats with outdoor access were reported to display behavioral or temperamental problems. However, there was no effect of the length of time a cat was left home alone. Having studied animal behavior at university level influenced the perception of some cat behaviors, but not the incidence of reporting perceived problems. Many owners did not believe that it was possible to prevent behavioral problems in cats by training (58.5%), and many owners thought that cats could manage independently on their own (66%). Attitudes like this can cause challenges in the owner–cat interactions. The perception of problems with cats will be influenced by factors relating to husbandry routines such as outdoor access, which in the future could help to implement recommendations for cat husbandry and care.

## 1. Introduction

With over 1.4 million cats in 20% of Swedish households, the domestic cat (*Felis silvestris catus*) is Sweden’s most numerous companion animal [1]. Still, there have been no studies investigating the owners’ attitudes or how cats are housed and how this influences their behavior and welfare. Even though the domestic cat (hereafter, cat) is such a common pet, both in Sweden and in other parts of the world (e.g., Europe and the U.S.) [2], knowledge about basic cat behavior and needs have previously been described as limited among both owners and veterinary professionals (e.g., [3,4]). A lack of knowledge on how to provide a suitable physical as well as social environment can result in cats experiencing fear and stress, subsequently leading to the development of undesired behaviors [5]. Undesired behaviors include any behavior deemed unwanted or unacceptable by the owner, independent of the potential impact on welfare. They are often multifactorial and complex issues for cat owners to solve [6]. This can be related to the owners’ lack of awareness about basic cat behavior [4,7] such as believing that cats act out of spite [8], or providing inconsistent husbandry and handling [9].

In the United States, Salman and colleagues [8] found that the main reason for relinquishing cats to shelters was due to undesired behaviors from the owners’ point of view. Heidenberger’s [10] survey in Germany found that half of the participating cat owners perceived behaviors they wished to change. House soiling, furniture scratching, feeding problems, and vocalization were reported as the most common problems. Cats with owners who were aware of basic cat behavior and environmental needs showed fewer reported undesired behaviors as perceived by the owners [11]. Similarly, owners who had read books or other materials relating to cat behavior were considered as protective factors against relinquishment, while having specific expectations of the relationship and the role a pet cat will play were seen as a risk factor for relinquishment [8].

The majority of cat owners seem to consider their pet as part of the family: 92% in the U.S. [12], 93% in a U.S. poll [13], and 60% in an Australian survey [14]. Concerning the relationship between cats and owners, in a study focusing on the cats’ quality of life (QoL), Adamelli et al. [15] found that 83.9% were said to express some sort of undesired behavior such as elimination problems. Furthermore, they found an effect of the owner features such as owner age, education, and number of family members on QoL. Looking at the association between housing (single- or multi-cat housing) and stress levels (urinary cortisol), factors positively correlated with stress included human density and the total available space in a household [16]. Provided environment as well as human factors seem to have a large influence on the welfare of cats.

The behavior of cats is influenced by a combination of genetics, early experience, and the provided environment [17], both social and physical. The opinions of owners differ with regard to several factors such as the provision of outdoor access or not for pet cats. It is still unclear how outdoor access influences cat behavior and the owner perception of undesired behavior. Indoor and outdoor cats are susceptible to different welfare issues, and confinement itself can have a negative effect on cats (reviewed by [18]). Issues with indoor housing only may involve difficulties in meeting some of the innate needs of cats [4]. However, some also argue that cats should be kept indoors for their own safety [19], and for the threat that they pose to wildlife [20]. Previous research also indicates that owner knowledge affects the quality of the environment provided for cats. 

### Aim

The overall purpose of the present study was to increase the understanding of cats and their owners by investigating how pet cats are housed and cared for in Sweden, and how this correlates with the owner characteristics. Specific aims were (i) to determine if the time a cat is left home alone is related to perceived behavioral or temperamental problems; (ii) to identify how cats’ access to the outdoors affects the perceived behavioral and temperamental problems; and (iii) to assess whether the experience of higher education and ethological knowledge influences the perception of cat behavior or behavioral/temperamental problems.

## 2. Material and Methods

### 2.1. Questionnaire Development

The web-based survey used in this study included a translation of a prototype of a survey developed by Prof. James Serpell and colleagues at the University of Pennsylvania [21]. The survey was developed with the aim of evaluating behavior and temperament in companion cats based on owner reports, and has been confirmed for construct validity. The version used in this study in Sweden comprised 21 sections and a total of 93 questionnaire items, each expressed as a 5-point ordinal rating scale that measures the frequency of occurrence of a wide variety of common cat behaviors.

The final survey included an additional 17 background questions regarding each cat’s health, history, and lifestyle, and the owner’s perceptions of the cat’s behavioral or temperamental problems. Five questions were also incorporated (relating to the owner’s education about animal behavior, previous cat experience, and the respondents’ views on cat sociability and behavior. We report the results from questions relating to the time the cats were left alone, cat lifestyle, perceived behavioral and temperamental problems, owner education, and agreement with claims about cats in relation to cat and human factors. The survey was translated in its full form, despite the fact that some of the original items included reference to actions that would not be permitted under Swedish animal welfare legislation (e.g., declawing and the age of weaning). This was explained to the participants in an introductory text. Survey questions were posed mostly as interval scale questions, ratio scale questions, and open free text question. Multiple cat households were asked to only answer for one cat, chosen randomly. The study was given ethical approval from the department of animal environment and health, where the authors are situated. However, no ethical approval was needed from the national ethical board, since no animals were involved in the study. The survey was created using Netigate © Netigate AB (www.netigate.se) and was available between 29 April 2014 and 1 July 2014.

### 2.2. Subject Recruitment

Data are based on a convenience sample. Requirements for participation stated that an owner should own at least one cat and have access to the Internet. Recruitment for the survey took place online via the Swedish University of Agricultural Sciences home page, different platforms for cat owners such as pet store webpages, cat shelter websites, and SVERAK (Swedish Cat Breeders Association). The survey was also distributed on social media (Facebook) where viewers were encouraged to share the link to the survey on their own social media outlets to increase the number of people reached (snowball sampling). Survey participants were informed on how the data were planned to be used, and that the results would be published anonymously. Four thousand two hundred and nine (4209) Swedish cat owners responded to the survey of which 77% (3253) were complete and valid.

### 2.3. Data Analysis

The data were transferred from Netigate and sorted in Microsoft ^®^ Excel ^®^ 2010, where the incomplete responses were excluded. Answers where the owners had not followed the instructions and answered for two cats or more were disregarded as incomplete.

Analyses were performed using Minitab ^®^ (Statistical software version 16.1.0 © 2010 Minitab LLC State college, Pennsylvania.). Caretakers and the cats’ general home environment and health status were summarized using descriptive statistics. Associations between the perceived behavioral or temperamental problems, outdoor access, and owner education were analyzed using chi-square tests, and effect size (φ) was rounded to the nearest decimal number. For analysis relating to outdoor access, all cats with some form of outdoor access (free, enclosed area, and supervised walks, i.e., on/off leash) were grouped into a single category.

## 3. Results

### 3.1. Descriptive Findings

Of the 3253 included cats, 57% were domestic short- or longhair, 28% were pure bred, and the rest were mixed breeds. Seven cats (0.22%) were declawed, a procedure that is illegal according to the Swedish Animal Welfare Act [22]. Forty-eight percent were female, 90% were neutered, and 34% had some form of free outdoor access. A demographic summary of the cats in this study can be found in Table 1.

Respondents belonged to households comprising a median of two persons (IQR 2, min 1, max 12) keeping a median of two cats (IQR 2, min 1 max 22). The majority of respondents (84%) had previously owned 1–2 (33%) or three or more (51%) cats. Ten percent had studied animal behavior at the university level, 7% were active breeders at the time of the survey, and 3% had previously been cat breeders. It was most common to leave the cat alone for 6–8 h/day (26%), followed by 4–6 h/day (20%). Fifteen percent of cats were left alone for more than 8 h per day.

The vast majority of respondents (99%) agreed or partially agreed to the statements that “cats need environmental enrichment” and that “my cat appreciates my company” (Table 2). Fifty-four percent of respondents agreed with the statement that “cats can be trained.” Only 41.5% believed that it is possible to remove undesired feline behaviors by training. Sixty-six percent agreed or partially agreed to the statement that “cats can manage on their own.” Data concerning the cat owners’ attitudes and views of cats are summarized in Table 2.

### 3.2. Owner Education

There was an association between having studied animal behavior at the university level and agreement with the statement that “cats misbehave out of spite”. If the respondent had studied animal behavior, there was a lower frequency of agreement with the statement than expected (χ^2^ = 38.016, *p* = 0.0001, N = 3253), φ = 0.1. There was no association between exposure to higher education and knowledge of animal behavior and reported perceived behavioral or temperamental problems (χ^2^ = 0.702 *p* = 0.402).

Of the respondents, 82% reported no perceived behavioral or temperamental problems. Of the respondents who did report undesired behaviors (*n* = 592, 18%), 536 respondents used the free text response to describe 710 different perceived behavioral or temperamental problems. The majority, 87%, of the 710 issues were connected to responses related to cat behavior (Figure 1). Of the respondents perceiving temperamental problems, 43 respondents experienced only problems with their cat’s temperament, seven moderate, and 36 minor problems (Table 3).

A chi-square test of independence was used to investigate the association between outdoor access and the owner perceived behavioral or temperamental problems. There was an association between outdoor access and perceived behavioral or temperamental problems χ^2^ (1, N = 3241) = 16.255, *p* = 0.0001, φ = 0.1. Cat owners keeping cats with outdoor access had a lower observed reporting of problems than expected by chance. No association was found between the time left alone and perceived behavioral or temperamental problems χ^2^ (6, N = 3253) = 10.76 *p* = 0.096, φ = 0.1.

## 4. Discussion

Over 3000 cat owners completed this survey on their pet cats’ behavior and the influence of husbandry and owner features. Most of the cat owners (82%) did not experience any behavioral or temperamental problems with their cats, and we can conclude that the cat owners in this investigation were likely to be experienced cat owners since many have had cats before (84%). The most common husbandry situation reported was indoor housing, in an apartment, with some form of outdoor access for the cat.

### 4.1. Knowledge and Attitudes

In total, 66% of respondents agreed (10%) or partially agreed (56%) with the statement that cats could manage on their own, without care from an owner. Concerning cat health, previous surveys have shown that 11% of cat owners believe that cats do not get sick, and if they do, 7% believe that they can care for themselves (7%) [23]. This statement, and the consequences it can have on the lack of care provided to cats, is very problematic from a welfare perspective. Previous studies investigating the life and welfare of free-roaming cats have found that only between 16% [24] and 25% [25] of kittens survive over the age of 6 months and only 9.5% over the age of 10 months [26]. The life expectancy of barn cats is between 3 and 5 years compared to owned cats, with a life expectancy of 12.1 years in the U.S. [27] and 14.1 years in the UK [28]. In Sweden, pet insurance data on purebred cats shows that the median life expectancy is over 12.5 years [29]. We would like to stress that this belief in cats being independent can be very problematic as it might result in the owners of cats being less prone to visit the veterinarian for preventive health care for their cat. In households with both cats and dogs, cats are taken to the veterinarian significantly less often than dogs, according to a U.S. study where 72% of cats were taken less than once a year [23].

### 4.2. Behavioral or Temperamental Problems in Relation to Husbandry

Of the specified behavioral and temperamental problems, 710 different descriptions, 87% described issues with the behavior of their cat. There are several terms in use in the literature for behaviors experienced by owners, and/or society, as unwanted or problematic, but there is consensus on clear definitions [30]. In this discussion, we distinguish behaviors that are considered unacceptable, unwanted, and/or problematic (undesired behaviors) by the owner by the potential effect it has on the cat’s welfare in accordance with [17]. Behaviors that are part of the normal behavioral repertoire of cats but are unacceptable by the owner, and/or society such as furniture scratching are referred to as problem behaviors. Problem behaviors only pose a risk for the cat’s welfare if the owner chooses to inhibit the expression or punish the cat. Behaviors that indicate that the cat is ill, and a clear welfare issue involved such as compulsive disorders or stereotypic behaviors are referred to as behavior problems.

Due to the low reporting of experiencing problems with behavior and temperament (18%) it is clear that cat owners in the present study might not be representative of the Swedish cat owning community. Studies from Denmark [31], Germany [10], the U.S. [32], and an international survey focusing on urinary house soiling [33] show that approximately 50.1%, 54.7%, 61%, and 53.9% of cats display undesired behavior(s) from the owner’s point of view, respectively. The difference between the present and previous published results could also be attributed to differences in the formulation of questions, the questionnaire design, or legislation and cultural differences in pet keeping between countries. However, Finka and colleagues [34] also found that 20% of respondents in UK reported undesired behaviors. Clearly, some behaviors that are considered as undesired by professionals, may not be considered as problematic by the owners and vice versa, depending on the owner expectations. Here, the owner personality might be a contributing factor, which was not investigated in the present study, but which has previously been investigated. For example, Finka and colleagues [34] discussed whether more neurotic owners in the UK might over report issues regarding their cat and its behavior.

Previous studies have reported that about half of the owners experiencing undesired problems with their cats’ behavior consulted their veterinarian about the issues [32], and few owners, 25–30%, [35], considered the behaviors problematic enough to seek professional help. However, owners were less likely to seek help that cost money such as working with an animal behaviorist, even though cat owners were more likely than dog owners to seek help by paying for a veterinary visit and discussing the problem [36]. Discussing undesired behaviors with a veterinarian is an option, and 99.7% of the responding veterinarians considered behavioral problems to be a veterinary responsibility [6]. Still, previous studies have found that many veterinarians do not have more knowledge about cat behavioral needs than the owners [3]. The lack of knowledge of veterinary professionals on cat behavior has also been suggested by the American Association of Feline Practitioners [4] and might be the reason why owners do not seek their help.

The majority of cats in this study were reported to be neutered (90%), which is slightly higher than previous reports (e.g., 79.8% in the U.S. [12], 81% in the U.S. [32], 86.1% in Denmark [31], and 78.7% in Germany [10]. Previous studies have reported a lower prevalence of undesired behaviors in neutered compared to intact cats [31], which might also be a contributing factor to the low incidence of reported behavioral and temperamental problems in this study.

The majority (76%) of responding veterinarians reported that at least 10% of euthanasia cases involving cats in Spain were due to undesired behaviors [6]. Urinary house soiling, one of the most commonly reported undesired behaviors (e.g., from 12.4% [31] to 53.9% [33]), is often not specified as inappropriate elimination or marking behavior, but both may indicate welfare problems for the cats. Inappropriate elimination may relate to underlying illness such as feline idiopathic cystitis (FIC), aversion to the litterbox (due to substrate, placement, or litterbox design), within household competition, stress, or maintenance. Marking behavior or spraying is mostly related to stress or fear and anxiety or a lack of more appropriate ways to perform marking behavior such as scratching at a post. Providing an adequate environment, suited to accommodate normal feline behavior, seems to be a factor in preventing undesired behaviors and thereby decreasing the risk of euthanasia. Informing cat owners about feline friendly environments such as providing an enriched environment including a safe space and the opportunity for cats to express normal behaviors should be a priority in the prevention of undesired behaviors and subsequent feline welfare problems.

We could not find a statistical relationship between having studied animal behavior at a university level and the reported behavioral and temperamental problems in owned cats. This was surprising, as previous studies have found that providing owners with behavioral advice can result in fewer undesired behaviors, at least in kittens [11]. Having read a book or materials about cat behavior can protect cats from relinquishment for which one of several risk factors are owner perceived undesired behavior [8]. However, our question was not directly about cat behavior but any behavior, and might therefore be too general to find a significant difference. Furthermore, the incidence of problem behavior was very low and a correlation with education could consequently be difficult to identify.

### 4.3. Outdoor Access

Of the respondents, 23% stated that their cat was indoors only, 34% had free outdoor access with an additional 22% having some form of outdoor access (e.g., enclosed area or walking on leash). This is comparable to previous reports in U.S. surveys of 37% [32] and 32% [12] of cats having some form of outdoor access and 64% [32] and 60% [12] being kept indoors only. In a Danish survey, utilizing a representative sample of the population, 69.1% of the cats had outdoor access and could roam freely by either being let out by the owner or using a cat-flap, and 22.4% were confined strictly indoors [31]. In a cross-sectional study from a veterinary referral center, cats acquired from shelters were reported to be kept indoors more often than cats acquired as strays [37]. In Sweden, many shelters require that the new owners agree to keep the cat indoors. The owner perceived undesired behaviors in cats showed that cats kept indoors only were more likely to be obese, exhibit pica (ingestion of non-nutritive substrates), and suffer from hairballs [38]. Cats without outdoor access were significantly more likely to show behavior problems than cats with outdoor access in a sample of cats admitted to a behavior service at a veterinary hospital [39]. In contrast, Strickler and Shull [32] found the support of a reversed relationship.

We found a significant relationship between providing outdoor access and owners reporting fewer behavior and temperamental problems. Our data suggest that outdoor cats are protected from undesired behaviors compared to cats kept only indoors. Still the number of cats housed strictly indoors is increasing throughout Europe and the practice is already common in the U.S., where veterinary associations such as the American Veterinary Medical Association recommend this for cats in urban and suburban locations. The finding that outdoor access can reduce the prevalence of certain undesired behaviors has also been previously reported [31].

Indoor cats (*n* = 577) spend a large proportion, 3 to 4 h (33.3%) and over 4 h (30.8%), of their day looking out the window at outdoor stimuli [40]. As cats have been reported to spend a large part of the day resting or sleeping, between 48% [41] and 80% [42], the opportunity to observe the outdoor environment should be seen as an important resource. During a study of the effect of available space on the behavior of indoor housed cats, Loberg and Lundmark [43] found that cats seldom shared resources with only one horizontal level (e.g., benches) with the exception of the bench in front of the window. In the same sense, Ellis [44] recommended increased complex visual stimulation (e.g., by providing access to windows for indoor-only cats) and cautioned that restricted or no outdoor access could induce negative emotional states in cats. Outdoor access has earlier been recommended as a way to reduce undesired behaviors by providing stimulation and reducing stress and boredom [45].

### 4.4. Time Left Alone

No association between the time a cat was normally left alone each day and owner perceived behavior and temperamental problems was found. Interestingly, only 15% of cats were left more than 8 h/day, which would be normal for someone working full-time in Sweden. A majority (63%) of respondents agreed with the statement that cats disliked being left alone and 91% agreed that their cat appreciated their company. Observations of the effect on cat behavior after separation from the owner showed that the time left alone did not affect cat behavior during the separation, but cats purred more and showed more body stretching at reunion with the owner after longer separations [46]. Even if cats originate from a solitary species, the North African wildcat, we would like to suggest that the domestication process has resulted in some changes in cat social behavior as the domestication process includes a selection pressure for social tolerance, both inter- and intra-specific. In a study of cat preferences, utilizing a free operant assessment, 50% of cats chose human interaction over food, play, and scent enrichment [47].

### 4.5. Environmental Enrichment

Of the respondents, 90.5% agreed with the statement that cats need environmental enrichment. There are several different definitions for environmental enrichment such as “improvement in the biological functioning of captive animals resulting from modifications to their environment” [48]. However, it has previously been described that many cat owners are unaware of their cats’ behavioral and environmental needs. Results from a Portuguese sample (*n* = 130) of owners of indoor-housed cats found two major issues with cat husbandry: the first was a lack of knowledge relating to feline welfare such as environmental enrichment, and the other was the lack of effort in implementing known enrichments [49].

Half of the respondents (54%) agreed with the statement that cats are trainable, and training cats can be part of enriching the environment. Cats respond well to positive reinforcement [50] and quickly connect behaviors that result in positive outcomes. Strickler and Shull [32] reported significantly fewer behavior problems in cats with owners that played with their cat for longer bouts (at least 5 min compared to 1 min). Studies of shelter cats deemed frustrated found that training cats to perform ‘high-five’ using positive reinforcement had positive effects on cat behavior and welfare, as seen in increased levels of immunoglobulins and a decrease in respiratory disease [50].

### 4.6. Challenges in Interpretation and Assessment

It should be noted that the presented data were based on a convenience sample, and not a representative sample, which is a limitation. Additionally, the data were collected in 2014, and even if there had been no major changes in the legislation implemented, the potential effects of the coronavirus pandemic on cat husbandry have not been investigated. An estimation of the proportion of purebred cats in Sweden is approximately 9% [51] and in our sample, 28% were purebred. Perhaps only dedicated owners took the time to participate in this survey, which could partly explain the low reporting of perceived undesired behaviors (18%). However, this is the first large demographic study of pet cats and their owners in Sweden, so it is not possible to compare the results with previous findings.

## 5. Conclusions

In our sample, there was a relation between the provided husbandry on the owner perceived cat behavior. In accordance with previous findings, we also found that providing outdoor access is related to a lower incidence of undesired behavior. It seems that even providing outdoor access in the form of walks on a leash has a positive effect on owner perceived cat behavior. This could be important findings as the perception of undesired behavior can have a negative effect on the cat–human relationship and has previously been related to the relinquishment of cats. There is also a risk that cats will be punished for displaying behaviors that are unacceptable or undesired by the owners. Even if not investigated here, there are data in support of the owners’ lack of knowledge about natural cat behavior. Consequently, some perceived undesired behaviors may actually be normal feline behaviors expressed in a way that the owners dislike, and does not necessarily indicate a welfare problem. Even if we did not find an effect of studies in animal behavior at the university level on the perceived behavioral and temperamental problem with one’s own cat, the studies did affect the perception of cat behavior. Respondents having studied animal behavior did not agree with the statement that cats misbehaved out of spite. Educating owners on normal feline behavior can be helpful in providing the optimal environment to decrease the risk of undesired behaviors, and through that, maintain a positive relationship and reduce the risk of abandonment or the relinquishment of cats.

## Figures and Tables

**Figure 1 animals-12-02664-f001:**
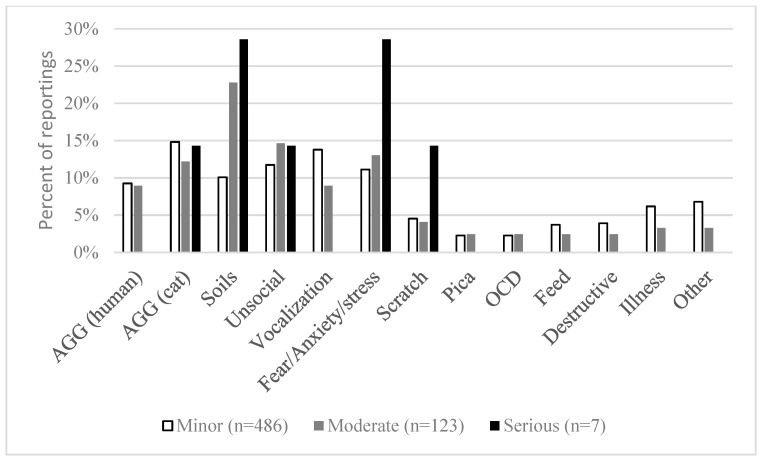
Distribution of the 616 reported perceived issues with behavior derived from the free text option for respondents (*n* = 549) stating that they experienced some form of problem with their cat’s behavior. AGG—aggressive toward familiar humans or cats; Soils—all forms of elimination problems; Unsocial—exhibiting fear and/or aggression toward familiar or unfamiliar people; Vocalization—undesired or excessive vocalization; Fear/Anxiety/Stress—described as exhibiting fear and/or anxiety and/or stress; Scratch—scratching furniture, walls, plants, etc.; Pica—ingestion of non-nutritive substances; OCD—obsessive/compulsive disorders; Feed—feed and feeding related problems; Destructive—cats described as waking owners during night and exhibiting destructive behaviors such as turning over items; Illness—issues relating to illness and pain; Other—all answers that could not be categorized.

**Table 1 animals-12-02664-t001:** A description of the cats involved in this study from the perspective of the owner.

		No.	%
Sex	Male	1704	52
	Female	1549	48
Neutered	Yes	2935	90
	No	318	10
Declawed *	Yes	7	0.2
	No	3246	99.8
Coat	Long-haired	337	11.6
	Semi-longhaired	777	23.9
	Short-haired	123	3.8
	Hairless	35	1
Breed type	Domestic shorthair	1628	50
	Domestic longhair	224	7
	Purebred	901	28
	Mix of purebreds	96	3
	Mix of domestic and purebred	404	12
Health problems	Yes	624	19
	No	2629	81
Lifestyle	Indoors only	877	27
	Indoors with access to outside enclosure	638	20
	Indoors with free outdoor access	1208	37
	Outdoors with no access to the house	24	1
	Lives in pen/stall/cage and has controlled access outdoors	1	-
	Indoors, walked on/off leash by owner	493	15
	Other **	12	-
Home environment	Apartment	1699	52.23
	House	1253	38.52
	Farm	247	7.59
	Professional breeder	42	1.29
	Cattery	0	-
	Shelter	2	0.06
	Other	10	0.31
Acquired from	Born in home, own breeding	214	6.58
	From a friend, relative or neighbor	905	27.82
	Veterinarian clinic	30	0.92
	Stray cat	318	9.78
	Shelter	475	14.6
	Professional breeder	891	27.39
	Cattery	5	0.15
	Pet store ***	0	-
	Other ****	415	12.76

* This procedure is illegal in Sweden. ** Category includes cats that did not fit into any predetermined category. *** Pet stores in Sweden are not allowed to sell cats or dogs. **** Category did not include a free-text alternative in the survey.

**Table 2 animals-12-02664-t002:** The respondent (*n* = 3253) agreement to different statements about cats.

Statement	Agree		Partially Agree		Disagree	
	No.	%	No.	%	No.	%
Cats need environmental enrichment	2948	90.5	284	8.5	21	1.0
Cats can manage on their own	335	10.0	1806	56.0	1112	34.0
My cat appreciates my company	2972	91.0	274	8.0	7	1.0
Cats are trainable	1758	54.0	1386	43.0	109	3.0
Cats act out of spite	268	8.0	811	25.0	2174	67.0
You can train away undesired behaviors	1352	41.5	1803	55.5	98	3.0
A cat dislikes being left alone	2053	63.0	1080	33.0	120	4.0

**Table 3 animals-12-02664-t003:** The distribution of the perceived temperamental problems, directly translated and suggested answers of the respondents (*n* = 43) reporting only issues with their cat’s temperament.

Temperamental Problem	No. of Respondents Severity		
	Minor	Moderate	Total
Cranky, moody, bad or short tempered	9	1	10
Attention seeking or demanding	5	3	8
Bored, restless, under-stimulated	6	1	7
Depressed, grieving	6	-	6
Jealous	2	1	3
“Wild”, boisterous	2	-	2
“False”	-	1	1
Dislikes indoors	1	-	1
Fertility issues	1	-	1
Curious	1	-	1
Dominant	1	-	1
Submissive	1	-	1
Persistent	1	-	1

## Data Availability

The data presented in this study are available on request from the corresponding author. The data are not publicly available due to the routines at the department.

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
