# Peer review of "Owner Perceived Behavior in Cats and the Influence of Husbandry Practices, Housing and Owner Attitudes in Sweden"

_animals, 2022, doi:10.3390/ani12192664_

Round 1

Reviewer 1 Report

Thanks for presenting this interesting study. The data were collected nearly 10 years ago, so could you please comment on whether you think the answers would be different if collected now.

Introduction: I would like the introduction to be reconsidered. I found the story being told jumped around and I wasn't sure exactly what I was meant to know by the end of it. QoL is introduced but not really tied to the behaviour being studied in this paper. Later you introduce the idea of behaviours that owners might not like but are normal cat behaviours and behaviours that are abnormal. Both these types of behaviours need to be introduced in the introduction and their connection to welfare and/or QoL made.

Cat behaviour as a cause of relinquishment  is introduced in Line 46 and then mentioned again in Line 76 -please pull these together.

Line 56 - 92% of cats where (the following percentages are given locations).

Aim: The three specific aims are good but do not seem to be a subset of the stated overall aim. Please explain further.

Line 152 - what does 'active' mean?

Discussion: In this study the question was asked about cats managing on their own whereas in the discussion it talks about cats not getting sick. These seem to be two different ideas. One seems to be about survival (maybe in the wild) and the other about health. Was the question being asked clear to the participants? Please clarify this section.

There are a few places where there isn't the correct subject and verb matching (plural/singular) e.g. Lines 217/8, 239, 249, 324.

In the discussion please make it clear when you are comparing your results with previous ones. For example, you refer to veterinarians' opinions - maybe tell us where this study occurred. 

Author Response

Dear reviewer, thank you for all valuable comments.

Best wishes

Reviewer 2 Report

The aim of the study and the methods are interesting and accurate. The main criticism is the presentation of the data and their statistical analysis. The authors would give a much higher value to their results, and would probably have more novel findings, by using multivariate analysis. Such approach would highlight some owners'and cats' profiles and could give the opportunity to sutdy the relationships between those. Such data would be very interesting for the prevention of mismanagement and human-cat conflicts.

As it is, the paper could be considered for publication, but with less value compared with including such analysis. 

Author Response

(The authors gave the same response as above.)

Reviewer 3 Report

Review of manuscript animals-1814900, Owner perceived behavior in cats and the influence of husbandry practices, housing and owner attitudes in Sweden.

The authors summarize the results of a survey addressing correlates of behavioral/temperamental problems in cats and the time that a cat is left alone, access to the outdoors, and whether the owner has taken a course on animal behavior.  As talked about in their discussion, some of these correlates have already been addressed in samples from other countries.

A lack of effect sizes for the χ2 tests makes it difficult to decide whether the manuscript offers a meaningful contribution to the literature or not.  If the effect sizes are small, then the manuscript probably does not make a substantial contribution to the literature given that many of the basic hypotheses have already been tested as cited in the discussion.  If the effect sizes are medium to large and other issues are adequately addressed, the manuscript would likely make a meaningful contribution to the literature.

More substantive issues:

If, as talked about in the discussion, some of these correlates of behavioral problems have already been addressed in the literature, what is the need to address them again?  The title of the manuscript suggests that something about the people of Sweden or the legal requirements of cat ownership in Sweden might be the answer.  Yet, these possible differences are not fully discussed in the introduction.  The manuscript needs to do a better job of explaining why the research is important given the numerous citations of similar results in the discussion.

This study is primarily descriptive with a few associations.  Were these associations hypothesized a-priori or were many possible associations tested and only those that were statistically significant reported?  If the potential associations were hypothesized a-priori, they should be explicitly stated in the introduction – the "Aim" section should be more fully developed.  If all possible associations were tested and only those that were statistically significant were reported, how did you protect yourself from Type I errors?

Effect sizes for the inferential tests must be reported given your large sample size.  With large samples it is easier to find statistically significant results, but those results might be of such a small magnitude that they are virtually meaningless.  Without the effect sizes, it is impossible to know whether the magnitudes of the effects are sufficiently large to be important or not.

The data were collected in 2014.  Have events in the intervening eight years possibly changed the results?  For example, many people adopted pets at the start of the pandemic, perhaps for the first time.  Would this impact your results?  Minimally, a short paragraph in the discussion should address this issue.

At first, section 4.1 seemed like two separate topics merged together – cats managing on their own and providing health care for sick cats.  I interpreted the question about whether cats can manage on their own without interference from an owner as meaning "on a day-to-day basis, do humans have to do more than feed and scoop for their cats?"  I wouldn't have considered a special case of the cat being sick as applying to this question.  I wonder how many of the respondents interpreted the question in the strict way that the authors seem to do – that humans have to do nothing for the cat and it will survive?   I think that it is likely that many respondents interpreted the question as I did given that only 10% agreed with the item and 56% partially agreed with the item.  The partial agreement could reflect that when not sick, cats don't require (but certainly may want) much, but there are situations (sick or injured) when they do need human intervention.  Given that, I'm not sure that the conclusion drawn in this paragraph is reasonable.

Because χ2 tests are sensitive to having the same individual in more than one cell, did you check your data to ensure that a given respondent did not respond more than once?

Minor issues:

Line 11:  "Very few" is subjective – personally, I would not consider 18% to be "very few".  Perhaps "Less than 1 in 5" or "Approximately 20%" would be more appropriate.

Line 12: Consider replacing the sentence with "Cats with access to the outdoors experienced fewer problems."

Lines 13, 284, 330: A lack of a statistically reliable association does not imply that the association does not exist.

Lines 14-15:  Awkward and imprecise.  "Owners who had taken an animal behavior course at university perceive cat behaviors (specify which behaviors) differently than those without such knowledge."

Line 16: Insert "Such" or "These" at the start of the sentence which is at the end of the line.

Line 26: Change "behavior" to "behaviors".  Change "however" to "but"                      

Lines 35-36: Typically, species names are italicized.

Lines 98-99: The appendixes were not included in the manuscript available for download for this reviewer.  Do you really have 43 appendixes to the manuscript?  If so, you need to cite the 38 appendixes that are not cited on lines 98 and 99.  If not, you need to clarify what you are talking about here.  Are "2, 3, 41, 42, and 43" the item/question numbers on the questionnaire?

Line 100:  Delete "In this paper". 

Lines 115, 377: "Data" is a plural noun and required a plural verb.  Ditto for "Requirements".

Line 116: Recruitment of participants for the survey took place on the listed platforms.  The survey per se was distributed on Netigate.

Tables 1, 2, 3:  "No" will be read as "the opposite of yes" by some readers instead of an abbreviation for "number".  Consider using "Number", "No." or "#" instead.

Table 1: A blank line between major categories (those in the first column) would improve the readability of the table.

Line 152: "7% were active" is ambiguous – does this mean that they are not sedentary or that they are currently cat breeders?

Line 152: Consider replacing "earlier" with "previously"

Lines 168, 170, 196, 198: It appears that the square symbol is not superscripted.  χ2 should be χ2

Lines 168, 196, 198.  "N" typically refers to the size of a population and "n" refers to the size of a sample.

Line 173:  Delete "a total of"

Line 201:  While true, the statement is misleading.  Over 4000 cat owners responded to the survey.  Over 3000 cat owners completed the survey.

Line 224: Delete "in total"

Author Response

(The authors gave the same response as above.)

Reviewer 4 Report

 The authors appear to have a bias against confining cats indoors, although they do reference the higher mortality of cats and birds  when cats have outdoor access. They briefly mention catios ( outdoor enclosures for cats) but do not mention the important feature of outdoor access - grass. Providing a cat garden  which can be easily grown in a flower pot or purchased, can give the cat access  The odds of aggression toward other cats were increased in female cats, cats with equal indoor/outdoor access compared those without outdoor access, OHanley et al.2021

https://www.sciencedirect.com/science/article/abs/pii/S0168159121000381?via%3Dihub

1)The main question was 

2)the topic is relevant

3) it adds to the subject area

4) the methodology-using videos provided by owners-is novel

5 the conclusions are consistent

Studying animal behavior at university may educate owners about the behavior of wild animals, birds, insects, and  possible mammals such as  baboons but is unlikely to  address domestic animal behavior.  This should be mentioned .

What are fertility issues? Presumably these are breeders.

Except for ground floor apartments apartment dwelling cats are not going to have  free outdoor access. This should be addressed in statistics especially because the majority of the cat owners lived in apartments

 Inappropriate elimination is not inappropriate for the cat. The problem is house soiling or eliminating outside the litterbox. Please use that terminology

 small grammatical point  fewer behavior problems not less. Fewer is for number; less is for amount

 The references are appropriate

Author Response

(The authors gave the same response as above.)

Round 2

Reviewer 2 Report

I have reviewed the revised version of this paper, and I have read the cover letter. Everything looks perfectly fine and I recommend this paper to be published in Animals.

Reviewer 3 Report

The revisions adequately address my concerns with the original manuscript.